# Predictive coding in balanced neural networks with noise, chaos and delays

**Jonathan Kadmon**
Department of Applied Physics
Stanford University,CA
kadmonj@stanford.edu

**Jonathan Timcheck**
Department of Physics
Stanford University,CA

**Surya Ganguli**
Department of Applied Physics
Stanford University, CA

## Abstract

Biological neural networks face a formidable task: performing reliable computations in the face of intrinsic stochasticity in individual neurons, imprecisely specified synaptic connectivity, and nonnegligible delays in synaptic transmission. A common approach to combatting such biological heterogeneity involves averaging over large redundant networks of $N$ neurons resulting in coding errors that decrease classically as $1/\sqrt{N}$. Recent work demonstrated a novel mechanism whereby recurrent spiking networks could efficiently encode dynamic stimuli, achieving a superclassical scaling in which coding errors decrease as $1/N$. This specific mechanism involved two key ideas: predictive coding, and a tight balance, or cancellation between strong feedforward inputs and strong recurrent feedback. However, the theoretical principles governing the efficacy of balanced predictive coding and its robustness to noise, synaptic weight heterogeneity and communication delays remain poorly understood. To discover such principles, we introduce an analytically tractable model of balanced predictive coding, in which the degree of balance and the degree of weight disorder can be dissociated unlike in previous balanced network models, and we develop a mean field theory of coding accuracy. Overall, our work provides and solves a general theoretical framework for dissecting the differential contributions neural noise, synaptic disorder, chaos, synaptic delays, and balance to the fidelity of predictive neural codes, reveals the fundamental role that balance plays in achieving superclassical scaling, and unifies previously disparate models in theoretical neuroscience.

## 1   Introduction

The early days of computing generated intense interest in how reliable computations could emerge from unreliable components, a question well articulated by von Neumann [1]. While the rise of digital technology largely circumvented this issue by making individual physical components highly reliable and fast, biological evolution, in the case of neural computation, had to directly face this problem. Indeed neural cortical firing patterns exhibit high levels of Poisson like temporal irregularity [2–4], external noisy inputs to a circuit can interfere with its operation, synaptic strengths are imprecisely specified in learning and development [5], and synapses themselves can be slow [6], resulting in non-negligible communication delays between neurons. Thus von Neumann's question still remains central in neuroscience [7]: how can neural circuits perform reliable computations when their underlying components, connectivity and inputs can be slow and subject to unpredictable fluctuations?

A conventional approach to this problem involves averaging over large redundant networks of $N$ neurons, resulting in coding and computation errors that decay as $O(1/\sqrt{N})$ as long as neural firing patterns are weakly correlated, due to the law of large numbers. However, can one do better? Recent

work [8] has constructed a recurrent network of spiking neurons that achieves *superclassical* error scaling, with the error decreasing as $O(1/N)$. Two key ideas underlying this network are the notions of predictive coding [9, 10] and balance [11]. In a sensory coding context, predictive coding refers to scenarios in which a neural circuit computes a prediction $\hat{x}(t)$ of some dynamic sensory input $x(t)$. Then a representation of the prediction error $\hat{x}(t) - x(t)$ can be employed for diverse purposes, including learning a causal model [12], cancellation of predictable sensory consequences of motor actions [13], mismatch between auditory and visual speech perception [14], or simply communicating surprises to downstream regions [15]. In [8] in particular, the prediction error was used to drive the dynamics of the recurrent spiking network through extremely strong negative feedback, thereby forcing the network prediction $\hat{x}(t)$ to track the sensory input $x(t)$. Furthermore, the gain $b$ of the negative feedback was proportional to network size $N$, resulting in a very *tight balance* or cancellation between strong feedforward drive due to the external input $bx(t)$ and recurrent negative feedback generated by the network prediction $-b\hat{x}(t)$.

A notion of balance has also played a prominent role in theoretical neuroscience in the context of a very different question: what mechanisms can generate the strong heterogeneity of observed biological firing patterns [2, 3] in the first place? [16, 17] demonstrated that disordered random connectivity itself can generate fluctuations in firing activity due to high dimensional chaos in neural circuits, without the need for additional injected noise. Moreover, recurrent networks in which each neuron receives strong excitation and strong inhibition, self-organize into a highly heterogenous balanced state [18, 19], where excitation and inhibition into each neuron is large and $O(\sqrt{N})$, but their difference cancels to $O(1)$ fluctuations which drive firing, a situation we term *classical balance*, in contrast to the *tight balance* of [8]. Given the empirically observed prevalence of highly heterogenous firing patterns in the brain, the dynamical operating regime of cortex, and in particular, the degree of excitation-inhibition balance involved (tight, classical, or something looser), remains a question of great interest [20, 21].

These two largely distinct strands of inquiry, namely exploiting tight balance to make predictive coding highly efficient, versus exploiting classical balance to explain the origins of neural variability itself, in the absence of any particular computations, raises several foundational questions. First, what is the relation between the chaotic networks of classical balance and the predictive coding networks of tight balance? What minimal degree of balance can generate superclassical scaling of error with network size? Indeed can we elucidate the fundamental role of balance in achieving superclassical scaling? Moreover, what is the efficacy of balanced predictive coding in the presence of noisy external inputs, chaos induced by additional weight disorder, or delays due to slow synaptic communication? While some of the latter issues have been explored numerically in predictive coding spiking networks [22, 23], a theoretical analysis of the interplay between balance, weight disorder, noise, chaos and delays in determining the fidelity of predictive coding has remained elusive due to the complexity of the network models involved. This lack of understanding of how multiple facets of biological variablity interact with each other in predictive coding represents a major gap in the theoretical literature, given the prevalence of predictive coding in many areas of theoretical neuroscience [9, 10].

We aim to fill this gap by introducing and analyzing a theoretically tractable neural network model of balanced predictive coding. Importantly, in our new model we can independently adjust the amounts of: balance employed in predictive coding, weight disorder leading to chaos, strength of noise, degree of delay, and the single neuron nonlinearity. In previous balanced network models for generating heterogeneity, the degree of chaos inducing weight disorder and the degree of excitation-inhibition balance were inextricably intertwined in the same random connectivity pattern [18]. Our model in contrast exhibits an interplay between low rank structured connectivity implementing balance, and high rank disordered connectivity inducing chaos, each with *independently* adjustable strengths. In general, how computation emerges from an interplay between structured and random connectivity has been a subject of recent interest in theoretical neuroscience [19, 24–26]. Here we show how structure and randomness interact by obtaining analytic insights into the efficacy of predictive coding, dissecting the individual contributions of balance, noise, weight disorder, chaos, delays and nonlinearity, in a model were all ingredients can coexist and be independently adjusted.

## 2 Linearly decodable neural codes in noisy nonlinear recurrent networks

Consider a noisy, nonlinear recurrent neural network of $N$ neurons with a dynamical firing rate vector given by $\mathbf{r}(t) \in \mathbb{R}^N$. We wish to encode a scalar dynamical variable $x(t)$ within the firing rate vector

$\mathbf{r}(t)$ such that it can be read out at any time $t$ by a simple *linear* decoder $\hat{x}(t) = \frac{1}{N}\mathbf{w}^T\mathbf{r}(t)$ where $\mathbf{w}$ is a *fixed* time-independent readout vector. The dynamical variable $x(t)$ could be thought of either as an input stimulus provided to the network, or as an efferent motor command generated internally by the network as an autonomous dynamical system [27]. For simplicity, in the main paper we focus on the case of stimulus encoding, and describe how our analysis can be generalized to autonomous signal generation in the Supplementary Material (SM) in a manner similar to previous studies of efficient coding of dynamical systems in spiking networks [8, 27, 28]. Also, while we focus on scalar stimuli in the main paper, our theory can be easily generalized to multidimensional stimuli (see SM).

We assume the nonlinear dynamics of the firing rate vector $\mathbf{r}(t)$ obeys standard circuit equations [29]

$$r_i(t) = \phi(h_i(t)), \qquad \text{and} \qquad \tau\dot{h}_i(t) = -h_i(t) + \sum_j J_{ij}r_j(t - d) + I_i(x(t)) + \sigma\xi_i(t). \quad (1)$$

Here $h_i(t)$ is the membrane potential of neuron $i$, $\phi$ is a neural nonlinearity that converts membrane potentials $h_i$ to output firing rates $r_i$, $\tau$ is the membrane time constant, $J_{ij}$ is the synaptic connectivity from neuron $j$ to $i$, $d$ is a synaptic communication delay, $I_i(x(t))$ is the stimulus driven input current to neuron $i$, and $\xi_i(t)$ is zero mean i.i.d Gaussian white noise current input with cross-correlation $\langle\xi_i(t)\xi_j(t')\rangle = \delta_{ij}\delta(t - t')$. Now the critical issue is, how do we choose the connectivity $J_{ij}$ and the stimulus driven current $I_i(x)$ so that the noisy nonlinear delay dynamics in (1) for $r_i(t)$ yields a simple linearly decodable neural code with a network estimate $\hat{x}(t) = \frac{1}{N}\sum_i w_i r_i(t)$ closely tracking the true stimulus $x(t)$? We generalize a proposal made in [8], that was proven to be optimal in the case of spiking neural networks with no delays, noise or weight disorder, by choosing

$$J_{ij} = g\mathcal{J}_{ij} - \frac{b}{N}w_i w_j, \qquad \text{and} \qquad I_i(x(t)) = bw_i x(t). \quad (2)$$

Here, $w_i$ are the components of the readout vector, which now appear both in the stimulus driven current $I_i$ and the connectivity $J_{ij}$ in a structured rank 1 manner. We also consider a random contribution $g\mathcal{J}_{ij}$ to synaptic strengths, modelling imprecision in connectivity. We take the structured connectivity to be random with $w_i$ chosen i.i.d from a distribution $\mathcal{P}(w)$ such that $w_i$ remains $O(1)$ for large $N$ with the norm of the vector concentrating at $\mathbf{w}^T\mathbf{w} = N$, while the random synaptic strengths $\mathcal{J}_{ij}$ are chosen to be i.i.d Gaussian variables with zero mean and variance $\frac{1}{N}$. Thus while the structured connectivity, which is $O(1/N)$, is much weaker than the random connectivity, which is $O(1/\sqrt{N})$, they each generate a comparable $O(1)$ contribution to the input current to any neuron (when $b$ is $O(1)$). Thus in this model, as $N \to \infty$, the input current to each neuron originates from 4 distinct sources, with 3 independently adjustable control strengths: input currents due to disordered connectivity ($g$), structured connectivity ($-b$), stimulus drive ($+b$), and noise ($\sigma$).

Interestingly, this model provides a simple and theoretically tractable instantiation of the principle of predictive coding of the stimulus through balance (See Fig. 1A). One can see this by inserting the connectivity in (2) into (1) and using the definition of the readout $\hat{x}(t) = \frac{1}{N}\sum_j w_j r_j(t)$ to obtain

$$\tau\dot{h}_i(t) = -h_i(t) + \sum_j g\mathcal{J}_{ij}r_j(t - d) + bw_i\left[x(t) - \hat{x}(t - d)\right] + \sigma\xi_i(t). \quad (3)$$

Thus the structured part of the recurrent connectivity implicitly computes a prediction of the stimulus $\hat{x}(t-d)$, which is then used to cancel the actual incoming stimulus $x(t)$, and the resulting coding error $x(t) - \hat{x}(t - d)$ drives membrane voltages $h_i$ in the readout direction $w_i$. The coefficient $b$ defines a level balance between positive feedforward stimulus drive, and negative feedback from the prediction computed by the structured connectivity. A key feature of our model is that, unlike in previous balanced network models [18, 19, 30], the degree of balance $b$ can be independently modulated relative to the degree of synaptic disorder, which here is controlled instead by $g$. Moreover, through different choices of scaling of $b$ with $N$, we can seamlessly interpolate between previously distinct regimes of balance, with $b = O(N)$ corresponding to tight balance [11], $b = O(\sqrt{N})$ corresponding to classical balance [19], and $b < O(\sqrt{N})$ corresponding to loose or no balance [21, 24, 31].

However, despite the prominent role of both balanced networks (e.g., [30, 32–35]) and predictive coding [9, 13–15] in theoretical neuroscience, to our knowledge, an analytic theory of the robustness of balanced predictive coding in the face of weight disorder, noise and delays in general nonlinear networks has not yet been developed. We take advantage of our simple analytically tractable model of balanced predictive coding in (1) and (2) to compute how the average error $\varepsilon^2 = \langle[x(t) - \hat{x}(t)]^2\rangle$

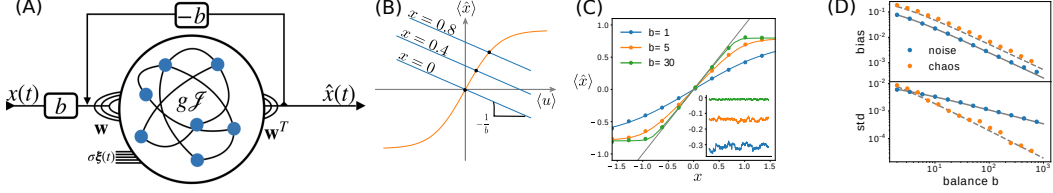

Figure 1: **(A)** A schematic view of a balanced predictive coding network. **(B)** Graphical solution method for mean field equations in (6) for $\phi = \tanh$. **(C)** The mean input-output transfer function $\langle \hat{x} \rangle$ as a function of $x$ obtained by solving (6) (solid curves) and numerical simulations of (3) (points) with $N = 1400$, $\sigma = 0.75$ and $g = d = 0$ for 3 values of $b$. Grey line marks $\langle \hat{x} \rangle = x$. The inset shows 3 corresponding examples of traces of $\hat{x}(t) - x$ when $x = 0.5$, demonstrating both bias ($y$-axis baseline) and fluctuations $\delta\hat{x}(t)$. **(D)** The decoder bias $\langle |x - \langle \hat{x} \rangle| \rangle$ (top) and standard deviation $\sqrt{\langle (\delta\hat{x})^2 \rangle}$ (bottom) as a function of balance $b$ for theory (curves) and simulations (points). $\sigma = 0.75$, $g = 0$ for noise (blue), $\sigma = 0$, $g = 1.6$ for chaos (orange). In both cases $N = 1400$ and $x = 0.2$. Balance $b$ yields power law suppression of variance with exponent $-1$ for noise and $-2$ for chaos.

of the neural code depends on various network properties. We work in an adiabatic limit in which the external stimulus $x(t)$ varies over a much longer time scale $T$ than either the membrane time constant $\tau$ or the delay $d$. Thus we can think of the stimulus $x(t)$ as effectively a constant $x$, and the squared error arises as the sum of a squared bias and a variance: $\varepsilon^2 = (\langle \hat{x} \rangle - x)^2 + \langle (\delta\hat{x})^2 \rangle$, where $\delta\hat{x} = \hat{x} - \langle \hat{x} \rangle$. The average $\langle \cdot \rangle$ can be thought of as an average over the realizations of noise $\xi_i$, or equivalently, a temporal average over an intermediate window of duration between that of the microscopic times scales of $\tau$ and $d$ and the macroscopic time scale $T$. Our goal in the following is to compute the bias and variance by computing the mean and variance of $\hat{x}(t)$ and its dependence on the strengths of noise $\sigma$, balance $b$, weight disorder $g$, delay $d$, and nonlinearity $\phi$.

## 3 A mean-field theory for bias and variance in a noisy neural code

We first consider the case of no weight disorder and delay ($g = d = 0$ in (3)), focusing on the interplay between balance $b$, nonlinearity $\phi$ and noise strength $\sigma$. To analyze these dynamics, we first decompose the membrane voltage vector $\mathbf{h}$ into two components, parallel and perpendicular to the readout vector $\mathbf{w}$, via $\mathbf{h}(t) = \mathbf{h}^{\|}(t) + \mathbf{h}^{\perp}(t)$ where $\mathbf{h}^{\|} = \mathcal{P}\mathbf{h}$ and $\mathbf{h}^{\perp} = (\mathbf{I} - \mathcal{P})\mathbf{h}$, and $\mathcal{P} = \frac{1}{N}\mathbf{w}\mathbf{w}^T$ is an orthogonal projection operator onto the direction of $\mathbf{w}$. Thus $\mathbf{h}^{\|}(t) = u(t)\mathbf{w}$ where $u(t) \equiv \frac{1}{N}\sum_{i=1}^{N} w_i h_i(t)$ and $\mathbf{h}^{\perp}$ obeys $\mathbf{w}^T\mathbf{h}^{\perp} = 0$. Now applying $\frac{1}{N}\mathbf{w}^T$ and $\mathbf{I} - \mathcal{P}$ to both sides of (3) we can decompose the dynamics into that of $u(t)$ and $h_i^{\perp}(t)$ respectively:

$$\tau\dot{u}(t) = -u(t) + b\left[x - \hat{x}(t)\right] + \sigma\xi^{\|}(t), \qquad \text{and} \qquad \tau\dot{h}_i^{\perp}(t) = -h_i^{\perp}(t) + \sigma\xi_i^{\perp}(t). \qquad (4)$$

The noise $\xi^{\|} = \frac{1}{N}\sum_{i=1}^{N} w_i\xi_i$ along the decoder direction now has diminished autocorrelation $\langle \xi^{\|}(t)\xi^{\|}(t') \rangle = \frac{1}{N}\delta(t - t')$, while the perpendicular noise components have autocorrelation $\langle \xi_i^{\perp}(t)\xi_j^{\perp}(t') \rangle = \delta_{ij}\delta(t - t')$ up to $O(1/N)$ corrections due to satisfying the constraint $\sum_i w_i\xi_i^{\perp} = 0$, which we can safely neglect. Thus in the large $N$ limit, the variables $h_i^{\perp}(t)$ undergo independent Ornstein Uhlenbeck (OU) processes each corresponding to leaky integration with time constant $\tau$ of white noise of variance $\sigma^2$, yielding an output with zero mean and temporal variance $\langle (h_i^{\perp})^2 \rangle = \frac{\sigma^2}{2\tau}$.

Next, in order to compute the temporal mean and variance of $\hat{x}(t)$, we decompose $u(t)$ into its temporal mean $\langle u \rangle$ and fluctuations $\delta u(t)$ about that mean via $u(t) = \langle u \rangle + \delta u(t)$. Inserting this decomposition into the dynamical equation for $u(t)$ in (4) and taking the temporal average $\langle \cdot \rangle$ of both sides, we obtain the relation $\langle u \rangle = b\left[x - \langle \hat{x} \rangle\right]$. We can obtain a second relation between $\langle \hat{x} \rangle$ and $\langle u \rangle$ by starting from the definition of $\hat{x}(t)$ and inserting the decompositions $h_i(t) = w_i u(t) + h_i^{\perp}(t)$ and $u(t) = \langle u \rangle + \delta u(t)$ to obtain $\hat{x}(t) = \frac{1}{N}\sum_{i=1}^{N} w_i\phi(h_i(t)) = \frac{1}{N}\sum_{i=1}^{N} w_i\phi\left(w_i\langle u \rangle + w_i\delta u(t) + h_i^{\perp}(t)\right)$. Now since $u(t)$ is driven by white noise $\xi^{\|}(t)$ of variance $O(1/N)$ in (4), we expect the fluctuations $\delta u(t)$ in the coding direction $\mathbf{w}$ to be of variance $O(1/N)$, and therefore much smaller than either the mean $\langle u \rangle$ or the perpendicular membrane voltages $h_i^{\perp}(t)$, both of $O(1)$, inside the argument of $\phi$.

Therefore we Taylor expand the nonlinearity $\phi$ about $\delta u(t) = 0$ to obtain to first order in $\delta u$:

$$\hat{x}(t) = \frac{1}{N}\sum_{i=1}^{N} w_i \phi\left(w_i\langle u\rangle + h_i^{\perp}(t)\right) + \frac{1}{N}\sum_{i=1}^{N} w_i^2 \phi'\left(w_i\langle u\rangle + h_i^{\perp}(t)\right)\delta u(t). \tag{5}$$

Now, taking the temporal average $\langle\cdot\rangle$ of both sides of this equation, we obtain, up to corrections of $O(\frac{1}{N})$, $\langle\hat{x}\rangle = \frac{1}{N}\sum_{i=1}^{N} w_i \langle\phi\left(w_i\langle u\rangle + h_i^{\perp}(t)\right)\rangle = \int \mathcal{D}z\, dw\, \mathcal{P}(w) w\phi(w\langle u\rangle + \frac{\sigma}{\sqrt{2\tau}}z)$. Here $\mathcal{P}(w)$ is the distribution of readout weights and $\mathcal{D}z = \frac{dz}{\sqrt{2\pi}}e^{-z^2/2}$ is the standard Gaussian measure. Thus we have obtained two equations for the two unknown means $\langle\hat{x}\rangle$ and $\langle u\rangle$:

$$\langle\hat{x}\rangle = x - \frac{\langle u\rangle}{b}, \qquad \text{and} \qquad \langle\hat{x}\rangle = \int \mathcal{D}z\, dw\, \mathcal{P}(w) w\phi(w\langle u\rangle + \frac{\sigma}{\sqrt{2\tau}}z). \tag{6}$$

The solutions to these equations can be viewed graphically (Fig. 1B). The first equation describes a straight line in the $\langle u\rangle$-$\langle\hat{x}\rangle$ plane with intercept $x$ and slope $-1/b$ (blue curves). The second equation behaves like a smoothed version of the nonlinearity $\phi$ (orange curve), and the intersection of these curves yields the solution. Thus as $b$ is increased, the slope of the line flattens, and the bias $|\langle\hat{x}\rangle - x|$ decreases, as long as $x$ lies in the dynamical range of the smoothed $\phi$. In general, the input-output behavior $x \to \langle\hat{x}\rangle$ is largely linear for all such values of $x$ at large $b$ (Fig. 1C). Our quantitative predictions for the bias are confirmed via numerical simulations in Fig. 1D, top. With knowledge of the nonlinearity $\phi$, degree of balance $b$, and noise level $\sigma$, one can theoretically compute the deterministic bias and remove it through the inverse map $\langle\hat{x}\rangle \to x$ when feasible. Therefore, we focus on the contribution of variance $\langle[\delta\hat{x}(t)]^2\rangle$ to coding error $\varepsilon$, which cannot be easily removed.

To compute the variance of $\delta\hat{x}$, we insert the decompositions $u(t) = \langle u\rangle + \delta u(t)$ and $\hat{x}(t) = \langle\hat{x}\rangle + \delta\hat{x}(t)$ into (4) and use the mean relation $-\langle u\rangle + b[x - \langle\hat{x}\rangle] = 0$ to extract a dynamic equation for the fluctuations $\tau\dot{\delta u}(t) = -\delta u(t) - b\delta\hat{x}(t) + \sigma\xi^{\|}(t)$. We then subtract $\langle\hat{x}\rangle$ from both sides of (5) to obtain the linearized relation $\delta\hat{x}(t) = \langle\phi'\rangle\delta u$ where $\langle\phi'\rangle \equiv \frac{1}{N}\sum_{i=1}^{N} w_i^2 \phi'\left(w_i\langle u\rangle + h_i^{\perp}(t)\right)$. Inserting this relation into $\dot{\delta u}(t)$ and replacing the sum over $i$ with integrals yields

$$\tau\dot{\delta u}(t) = -\delta u(t) - b\langle\phi'\rangle\delta u(t) + \sigma\xi^{\|}(t) \quad \text{where} \quad \langle\phi'\rangle = \int \mathcal{D}z\, dw\, \mathcal{P}(w) w^2 \phi'(w\langle u\rangle + \frac{\sigma}{\sqrt{2\tau}}z). \tag{7}$$

This constitutes a dynamic mean field equation for the membrane voltage fluctuations $\delta u(t)$ in the coding direction $\mathbf{w}$, where the average gain of the nonlinearity $\langle\phi'\rangle$ across neurons multiplicatively modifies the negative feedback due to balance $b$. Again, this is an OU process like that of $h_i^{\perp}$ in (4) except with a faster effective time constant $\tau_{\text{eff}} = \frac{\tau}{1+b\langle\phi'\rangle}$ and a smaller input noise variance $\sigma_{\text{eff}}^2 = \frac{\sigma^2}{N(1+b\langle\phi'\rangle)^2}$ yielding a diminished variance $\langle(\delta u(t))^2\rangle = \frac{\sigma^2}{2\tau N(1+b\langle\phi'\rangle)}$ both due to effective negative feedback, and averaging over the decoder direction $\mathbf{w}$. Note the fluctuations of $\delta u$ are indeed $O(1/N)$ making our initial assumption self-consistent. Finally, the variance of the readout fluctuations follows from squaring and averaging both sides of $\delta\hat{x}(t) = \langle\phi'\rangle\delta u(t)$, yielding

$$\langle(\delta u(t))^2\rangle = \frac{\sigma^2}{2\tau N(1 + b\langle\phi'\rangle)}, \qquad \text{and} \qquad \langle(\delta\hat{x}(t))^2\rangle = \frac{\langle\phi'\rangle^2\sigma^2}{2\tau N(1 + b\langle\phi'\rangle)}. \tag{8}$$

Taken together, the equations (6), (7) and (8) constitute a complete mean field theory of how the first and second order statistics of the projection of the membrane voltages and firing rates onto the decoder direction $\mathbf{w}$, i.e. $u(t) = \frac{1}{N}\sum_i w_i h_i(t)$ and $\hat{x}(t) = \frac{1}{N}\sum_i w_i r_i(t)$ respectively, depend on the balance $b$ and noise $\sigma$, in the large $N$ limit. We compare the theoretically predicted decoder bias $\langle\hat{x}\rangle - x$ and variance $\langle(\delta\hat{x})^2\rangle$ with numerical experiments, obtaining an excellent match (see Fig. 1D and Figures below). We find that the standard deviation of the decoder output scales as $O(1/b\sqrt{N})$. This reveals a fundamental necessity of strong balance, in which $b$ must scale as $N^\chi$ for some $\chi > 0$, to achieve superclassical scaling with decoder error falling off faster than $O(1/\sqrt{N})$. In particular, error scaling of $O(1/N)$, as reported in [8, 11, 23], is possible when $\chi = 1$, for which individual weights are $O(1)$ and do not scale with network size. In Section 5 we present an important limitation on the possible scaling.

## 4 The interplay between balance and chaos induced by weight disorder

We next consider the effects of weight disorder alone, with no noise or delays ($g$ nonzero but $\sigma = d = 0$ in (3)). This network has been shown to exhibit a dynamical phase transition from being a fixed point attractor when $g \leq g_c$ to chaotic evolution induced by large weight disorder for $g \geq g_c$ [16]. The critical transition point $g_c$ depends on the nonlinearity $\phi$ and strength of inputs $x$. Roughly, higher nonlinear gains $\phi'(x)$ promote chaos by reducing $g_c$. However, $g_c$ does not depend on the degree of balance where chaos and balance coexist [36, 37]. For $g \leq g_c$, there are no temporal fluctuations, so the only source of error is bias, which is computable and therefore can be removed. Thus we focus on the chaotic regime $g \geq g_c$ in which the amplitude of chaotic fluctuations of membrane voltages $h_i(t)$ increases with $g - g_c$ [36]. In essence, the recurrent input $g\eta_i(t) \equiv g \sum_j \mathcal{J}_{ij} \phi(h_i(t))$ due to the random connectivity $\mathcal{J}$ acts like a source of chaotic noise, analogous to the stochastic noise source $\sigma \xi_i(t)$ studied in Sec. 3. A major difference however is that while the stochastic noise source is white across both neurons and time, with cross correlation $\langle \xi_i(t) \xi_j(t') \rangle = \delta_{ij} \delta(t - t')$, the chaotic noise is, up to $O(1/\sqrt{N})$ corrections, white across neurons, but not across time, with cross correlation $\langle \eta_i(t) \eta_j(t') \rangle = \delta_{ij} q(t - t')$. For chaotic models, the temporal autocorrelation function $q(t - t')$ must be solved self-consistently [38, 39], and within the chaotic regime it decays to a constant value on a time scale close to the membrane time constant $\tau$.

While the full solution for the chaotic system is highly involved (see SM for comments on the derivation), we can describe the main quantitative effects of chaos on predictive coding error through an exceedingly simple derivation, which we give here. Basically, we can account for the chaotic fluctuations simply by replacing the white noise $\sigma \xi_i(t)$ in (3) by colored noise $g\eta_i(t)$ with temporal autocorrelation $q(t - t') = \exp(-|t - t'|/2\tau)$, which qualitatively matches the typical self-consistent solution to $q(t - t')$ in the chaotic regime. While this simplification does not describe the spatial structure of the chaos, which resides on a low-dimensional chaotic attractor [40], it does capture the temporal structure of the chaos which, as we see next, primarily determines the error of balanced predictive coding. We then follow the noise based derivation in Sec. 3. The analog of (7) becomes

$$\tau \delta \dot{u}(t) = -\delta u(t) - b\langle \phi' \rangle \delta u(t) + g\eta^{\|}(t) \quad \text{where} \quad \langle \eta^{\|}(t) \eta^{\|}(t') \rangle = \frac{1}{N} \exp(-|t - t'|/2\tau). \quad (9)$$

Thus the fluctuations $\delta u$ of membrane voltages $h_i(t)$ in the decoder direction $\mathbf{w}$ are well approximated by a leaky integrator with negative feedback proportional to $b\langle \phi' \rangle$ driven by colored noise, which is a stochastic ODE that is well understood [41]. Importantly, when the auto-correlation time of the driving noise equals the membrane time constant, as in this case, the variance is given by (see SM) $\langle \delta u^2 \rangle \approx \frac{g^2}{2N\langle \phi' \rangle^2 b^2}$, yielding a decoder variance

$$\langle \delta \hat{x}^2 \rangle \approx \frac{\langle \phi'^2 \rangle g^2}{2N\langle \phi' \rangle b^2 N}.$$

This should be compared to the decoder variance in (8) in the case of white noise, which instead scales as $O(\frac{\sigma^2}{bN})$. A rough intuition for the difference between chaos and noise can be obtained by considering the Fourier decomposition of the dynamics. In the case of colored noise, the power of the fluctuations is concentrated at low frequencies, while for white noise it is evenly distributed across the spectrum. Increasing $b$ has two opposing effects: attenuating the fluctuations on the one hand, and allowing higher noise frequency through the synaptic low-pass filter on the other. In the case of colored noise, the latter affect is negligible due to decaying power spectrum. Notably, $\langle \delta u^2 \rangle$ scales with the balance $b$ exactly as the inverse square $1/b^2$, and is a result of the exact match between the time-constant of the noise autocorrelation function $q(t - t')$ and of the dynamics in (9), which are both equal to $\tau$ (see SM for details). Thus our analysis reveals the important prediction that balance much more effectively suppresses decoder variance due to chaos versus noise, with a power law decay exponent in $b$ that *doubles* when going from noise to chaos. We verify this important prediction in Fig. 1D.

## 5 The role of delays, balance and noise in the onset of oscillatory instability

In the previous two sections we have seen that increasing balance $b$ always suppresses decoder variance $\langle \delta \hat{x}^2 \rangle$, for fluctuations induced both by noise and chaos. We now consider the case of a nonzero synaptic communication delay $d$, focusing first on the case of noise and no chaos (i.e.

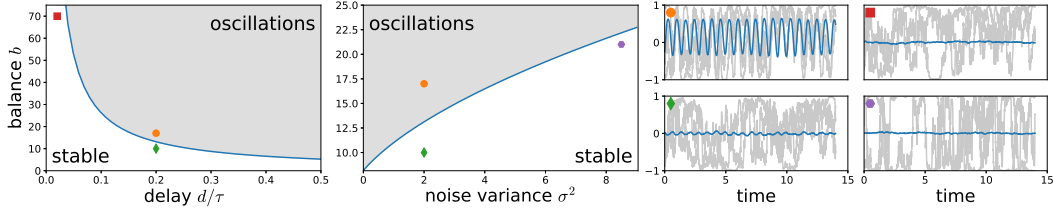

Figure 2: Dynamical phases in the presence of delays, balance and noise ($g = 0$, $x = 0.2$). Left: The critical balance $b_c$ (blue curve) as a function of the delay (with $\sigma^2 = 2$) obtained by solving for $\tilde{b}_c$ in (11) and dividing by $\langle \phi' \rangle$ in (7). Center: The critical balance $b_c$ as a function of noise $\sigma$ for fixed delay ($d/\tau = 0.15$). Right: sample firing rates $r_i(t)$ (grey) from simulations of (3) with $N = 1000$, with parameters corresponding to points in the left two panels, and the decoder trajectory $\hat{x}(t)$ (blue).

$d, \sigma > 0$ and $g = 0$ in (3)). In this setting, the entire derivation of Sec. 3 follows without modification until the analysis of membrane voltage fluctuation dynamics $\delta u(t)$ along the decoder direction $\mathbf{w}$ in (7). With a nonzero delay $d$, the dynamics of $\delta u(t)$ in (7) is modified to

$$\tau \delta \dot{u}(t) = -\delta u(t) - b\langle \phi' \rangle \delta u(t - d) + \sigma \xi^{\parallel}(t). \tag{10}$$

This corresponds to a delay differential equation [42]. We first consider its properties in the absence of noise input. First, for either zero balance $b$ or zero delay $d$, the dynamics has a stable fixed point at $\delta u = 0$. However, if either the delay $d$ is increased at fixed $b$, or the negative feedback $b$ is increased at fixed delay $d$, the combination of strong negative feedback $b$ and long delay $d$ can trigger an oscillatory instability. To detect this instability, we search for complex exponential solutions to (10) of the form $\delta u(t) = e^{zt}$ where the complex frequency $z = \gamma + i\omega$. These solutions correspond to stable damped oscillations at frequency $\omega$ if $\gamma < 0$, or unstable diverging oscillations if $\gamma > 0$. Inserting $\delta u(t) = e^{zt}$ into (10) yields a constraint on $z$ through the characteristic equation $G(z) = z\tau + 1 + \tilde{b}e^{-zd} = 0$ where $\tilde{b} \equiv b\langle \phi' \rangle$ is the *effective* negative feedback taking into account the average nonlinear gain $\langle \phi' \rangle$ in (7). At zero delay $d$, it has a solution $z = -(1 + b)/\tau$ indicating damped exponential approach to the fixed point $\delta u = 0$.

However, for a fixed delay $d$, as one increases the negative feedback $\tilde{b}$, the solutions $z$ to $G(z) = 0$ move in the left half of the complex plane with negative real part $\gamma < 0$ towards the imaginary axis with $\gamma = 0$. Let $\tilde{b}_c$ be the smallest, or critical value of $\tilde{b}$ for which $G(z) = 0$ first acquires solutions on the imaginary axis, indicating the onset of oscillatory instability for any $\tilde{b} \geq \tilde{b}_c$. We can find $\tilde{b}_c$ by searching for solutions of the form $G(i\omega_c) = 0$. The real and imaginary parts of this complex equation yield two real equations: $\tilde{b}_c \cos(\omega_c d) + 1 = 0$ and $\tilde{b}_c \sin(\omega_c d) - \omega_c \tau = 0$. Here, $\omega_c$ is the frequency of unstable oscillations at onset, when $\tilde{b}$ approaches $\tilde{b}_c$ from below. Solving for $\tilde{b}_c$ yields

$$\frac{d}{\tau} = \arccos(-1/\tilde{b}_c)/\sqrt{\tilde{b}_c^2 - 1}. \tag{11}$$

Thus the maximal stable negative feedback $\tilde{b}_c$ is a function only of the relative delay $d/\tau$. Indeed $\tilde{b}_c$ is a decreasing function of $d$, indicating the longer the delay, the weaker the negative feedback must be to avoid oscillatory instabilities. Beyond the linear oscillatory instability, with $\tilde{b} \geq \tilde{b}_c$, each neuron $i$ oscillates with amplitude proportional to $w_i$, stabilized by nonlinear saturation due to $\phi$.

Importantly, the critical balance $b_c = \tilde{b}_c / \langle \phi' \rangle$ depends on the average gain of the nonlinearity $\langle \phi' \rangle$, which in turn depends on the degree of noise $\sigma$ through (7). Increasing $\sigma$ spreads out the distribution of membrane voltages $h_i(t)$ across neurons $i$. For typical saturating nonlinearities, this *increased* spread in membrane voltages leads to a *decreased* average nonlinear gain, which in turn *raises* the critical balance level $b_c$, thereby allowing stronger negative feedback $b$ without triggering oscillatory instabilities. Essentially, longer delays promote synchrony, while noise suppresses it, at any fixed balance. The predicted phase boundary between stable noise suppression and oscillatory amplification in the simultaneous presence of noise, delays and balance is verified in simulations (Fig. 2). .

# 6 An optimal level of balance in the face of noise, chaos and delays

We now examine how the presence of the oscillatory instability of the previous section impacts the nature of optimal predictive coding, by considering how the delay dynamical system in (10) responds to the noise source $\sigma \xi^{\parallel}(t)$ in the stable regime, with $\tilde{b} \leq \tilde{b}_c$. We can understand the response in the frequency domain (see SM for detailed derivation). The power spectrum $\Delta(\omega)$ at frequency $\omega$ of the fluctuating time series $\delta u(t)$ can be written in terms of the characteristic function $G(z)$ as $\Delta(\omega) = [G(i\omega)G^*(i\omega)]^{-1}\sigma^2$, and the total variance is given by $\langle \delta u^2 \rangle = \int_{-\infty}^{\infty} d\omega \Delta(\omega)$. Now as $b$ approaches $b_c$ from below, the response power $\Delta(\omega_c)$ at the critical resonant frequency $\omega_c$ increases, since $G(i\omega_c) = 0$ when $b = b_c$. However, the power $\Delta(\omega)$ at non-resonant frequencies $\omega$ far from $\omega_c$ is suppressed by increasing $b$. Indeed the total variance of both $\langle \delta u^2 \rangle$ and $\langle \delta \hat{x}^2 \rangle$ can be approximated by the sum of the power in the nonresonant frequencies, calculated above in (8), and the power at the resonant frequency $\Delta(\omega_c)$, yielding (see SM)

$$\langle \delta \hat{x}^2 \rangle = \frac{\sigma^2 \langle \phi' \rangle^2}{2\tau N} \left( \frac{1}{1 + \tilde{b}} + \frac{1}{\tilde{b}_c - \tilde{b}} \right). \tag{12}$$

This expression exhibits a tradeoff: increasing $b$ attenuates the first term by suppressing non-resonant input noise frequencies, but increases the second term by amplifying resonant noise frequencies. Intriguingly, this fundamental tradeoff sets an optimal level of balance that minimizes decoder variance (Fig. 3). Indeed minimizing (12) yields an optimal balance $\tilde{b}_{opt} = \frac{1}{2}\tilde{b}_c$ (note $\langle \phi' \rangle$ does not depend on $b$ to leading order in $1/\sqrt{N}$). The resultant minimal error, $\varepsilon_{min}^2 = \langle \delta \hat{x}^2 \rangle$ as a function of the delay is shown in Fig. 3. For small delays $d \ll \tau$, the asymptotic expansion of (11) yields $\tilde{b}_c \approx \pi\tau/2d$, and so the error increases initially as the square-root of the delay and is given by

$$\varepsilon_{min} = 2\sigma \langle \phi' \rangle \sqrt{\frac{d}{N\tau\pi}}. \tag{13}$$

The expression for minimal error in (13) implies that in order to achieve error scaling of $O(1/N)$, as obtained in [8, 11, 23], the delay must scale as $d \sim \tau/N$. In practice, both the time constant $\tau$ and delay $d$ are fundamental properties of the system. Thus, error scaling of $O(1/N)$ can only be achieved in networks smaller than $N < d/\tau$.

**Weight disorder, chaos and delays.** Delays do not change the statistics of chaotic fluctuations, since the mean-field equations are stationary, and fluctuations at times $t$ and $t - d$ are equivalent. Moreover, the maximal critical balance $\tilde{b}_c$ does not depend on the fluctuations and is still given by (11). Below critically $b < b_c$ and for small delays $d \ll \tau$, resonant amplification at frequency $\omega_c$ plays less of a role in the case of chaos, since $\omega_c \propto 1/d$ and the power spectrum of chaotic fluctuations is exponentially suppressed at frequencies $\omega \gg 1/\tau$. Without a strong tradeoff between nonresonant suppression and resonant amplification, the optimal balance $b_{opt}$ for chaos is close to the maximal balance $b_c$, with a minimal decoder standard deviation that scales as $\varepsilon_{min} \propto 1/b_c$. For small delays where $b_c \sim \tau/d$, the minimal deviation scales as: $\varepsilon_{min} \sim d/\tau$. Our predicted scaling of optimal balance and deviation with delay in the case of chaos is confirmed in simulations (Fig. 3).

# 7 Discussion

In summary we have introduced a theoretically tractable nonlinear neural circuit model of predictive coding, and analytically derived many relations between coding accuracy and balance, noise, weight disorder, chaos, delays, and nonlinearities. We find: (1) strong balance is a key requirement for superclassical error scaling with network size; (2) without delays, increasing balance always suppresses errors via powers laws with different exponents (-1 for noise, -2 for chaos); (3) delays yield an oscillatory instability and a tradeoff between noise suppression and resonant amplification; (4) this tradeoff sets a maximal critical balance level which decreases with delay; (5) noise or chaos can increase this maximal level by promoting desynchronization; (6) the competition between noise suppression and resonant amplification sets an optimal balance level that is half the maximal level in the case of noise; (7) but is close to the maximal level in the case of chaos for small delays, because the slow chaos has small power at the high resonant frequency; (8) the optimal decoder error rises as a power law with delay (with exponent 1/2 for noise and 1 for chaos). Also, our model

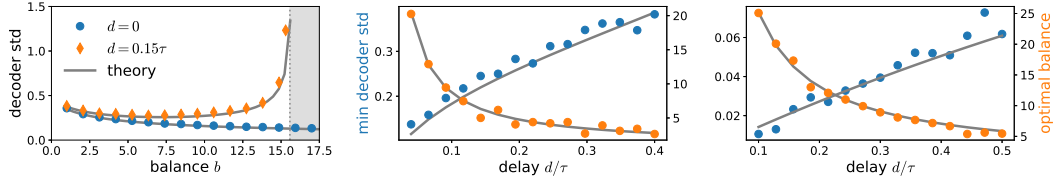

Figure 3: Optimally balanced network with delay, $\phi = \tanh$ and $x = 0$. Points reflect simulations of (3) with $N = 1400$ and curves reflect theory. Left: Decoder standard deviation ($\sqrt{\langle(\delta\hat{x})^2\rangle} \times \sqrt{N}$) as a function of balance $b$ with $\sigma = 0.75$. For $d = 0$ this deviation decreases monotonically with $b$ as predicted by (8). With nonzero $d$ this deviation exhibits a tradeoff between noise suppression and resonant amplification as predicted by (12), with strong global oscillations triggered at $b \geq b_c$ (grey region), as predicted by (11). The optimal $b$ occurs at $b_{opt} = b_c/2$ (see text). Center: optimal decoder standard deviation (blue) and $b_{opt}$ (orange) as a function of delay, given by (13) with $\sigma = 0.75$. Asymptotically, the error increases as $\sqrt{d/\tau}$. Right: Same as center but with deterministic chaos ($g = 1.6$, $\sigma = 0$). Theory curves are calculated via colored noise (see Sec. 4).

unifies a variety of perspectives in theoretical neuroscience, spanning classical synaptic balance [18, 30, 43–46], efficient coding in tight balance [8, 47], the interplay of structured and random connectivity in computation [19, 24, 25, 48, 49], the relation between oscillations and delays in neural networks [50–52] and predictive coding [9, 11]. Moreover, the mean-field theory developed here can be extended to spiking neurons with strong recurrent balance and delays [53], analytically explaining relations between delays, coding and oscillations observed in simulations but previously not understood [22, 23].

## Acknowledgments

JK thanks the Swartz Foundation for Theoretical Neuroscience for funding; JT thanks the National Science Foundation for funding. SG thanks the Simons and James S. McDonnell Foundations and an NSF Career award for funding. We thank ID Landau and H Sompolinsky for fruitful discussions.

## Broader impact

Our work is primarily theoretical and is designed to elucidate fundamental phenomena in nonlinear neural circuit computation. Such a scientific understanding, may, in the long term, lead to more robust technology.

## Funding disclosure

All authors are funded as detailed in the Acknowledgements Section above. None of the authors have competing interests.

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
