[Supplementary Material]

# Predictive coding in balanced neural networks with noise, chaos and delays

# Supplementary Material

**Jonathan Kadmon**
Department of Applied Physics
Stanford University,CA
kadmonj@stanford.edu

**Jonathan Timcheck**
Department of Physics
Stanford University,CA

**Surya Ganguli**
Department of Applied Physics
Stanford University, CA

## Contents

## 1 Encoding an autonomous dynamical system

In the main text, we considered a case where a neural network encodes a scalar input signal $x(t)$ in its dynamics. This simple example corresponds to the circuit acting as an autoencoder. The model is instructive, and allows us to rigorously study the effects of noise, weight disorder, and delays on the coding performance. In the simple case of an autoencoder, the desired output is explicitly provided to the network through the feedforward inputs. In a more general setting, the desired output of the network may be a complex spatiotemporal transformation of its input. The input-output transformation reflects the processing executed by the neural circuit, the details of which depends on the specific computation implemented. In this section, we show that the mechanism by which strong synaptic balance enables high-fidelity computations is general, and does not depend on origins of the signal. More precisely, we will show that the network can encode a linear dynamical system and that the resulting circuit equations obey similar balance rules as studied in the main text. Our assumption here is that the computed task can be written in terms of an autonomous linear dynamical system. Below we will also argue that this can be extended to other, nonlinear autonomous dynamical systems.

Our derivations follow closely those first suggested by [1] for spiking networks with integrate-and-fire dynamics. Here, we generalize the derivation to rate-based networks with arbitrary local nonlinear transfer functions. We explicitly show that the same ideas introduced in the theory of efficient coding in spiking neural networks [1, 2] apply to continuous firing-rate models. Furthermore, we emphasize that the crucial component allowing the network to encode an arbitrary linear dynamical system is a decoder that introduces an additional time scale.

## 1.1 Latent dynamical system

Consider again a network of $N$ nonlinear neurons. The output of each neuron is given by a nonlinear transformation $\phi$ of its input. We wish to implement the arbitrary linear dynamics for the latent vector $\mathbf{x}(t) \in \mathbb{R}^M$,

$$\tau_x \dot{\mathbf{x}}(t) = \mathbf{A}\mathbf{x}(t) + \mathbf{x}_0(t). \tag{S1}$$

Here, $\mathbf{x}_0(t) \in \mathbb{R}^M$ is the input, e.g., from external stimuli, $\mathbf{A}$ is an arbitrary state transition matrix, and $\tau_x$ is the timescale. We assume that $N \gg M$; this is a fundamental assumption and it is needed in order to obtain the statistical benefits of distributed coding. We refer to $\mathbf{x}(t)$ as the *latent* variable as it is not explicitly provided to the network, and its state is updated internally in the network.

The time scale of the dynamics $\tau_x$ can be very different than the microscopic timescale of the membrane potential, $\tau$. In general, we expect the dynamical time scale of interest to be much longer than membrane potential $\tau_x \gg \tau$; the choice of slow dynamics is equivalent to the adiabatic limit used in the main text. We also note that inputs of arbitrary dimensionality $M'$ can be fed into dynamics of this form, simply by multiplying by an $M \times M'$ input matrix to correct the dimension.

As in the main text, a linear readout provides an estimator for the encoded variable $\hat{\mathbf{x}}(t)$. Unlike the autoencoder model, the estimate is of the latent variable $\mathbf{x}(t)$, and not of the direct input to the network $\mathbf{x}_0(t)$. Importantly, to realize a dynamical system, we need to introduce a timescale relevant for the encoded system $\tau_x$. This timescale can be introduced through the readout,

$$\tau_r \dot{\mathbf{r}}(t) = -\mathbf{r}(t) + \phi(\mathbf{h}(t)). \tag{S2}$$

Here $\mathbf{r}(t)$ is a smoothed version of the activity $\phi(\mathbf{h}(t))$ with a linear low-pass filter. The slower dynamics of the readout provides the network with the necessary memory to implement the dynamics at slow time scales, even when the microscopic dynamics is fast and $\tau \ll \tau_x$. For brevity of our derivations, we let the readout time scale be $\tau_r = \tau_x$. In general, the readout and dynamics can have different time constants. However, if the readout is too slow it will not capture the high frequencies in the dynamics. On the other hand, if the readout timescale is too fast it will not have the necessary memory to implement slow dynamics. Thus, the readout timescale needs to comparable to that of the latent dynamics. Any finite differences can be incorporated into the circuit equations. For an autoencoder, which has no latent dynamics, there is no need for introducing a slow timescale in the decoder. In this case we take the limit $\tau_r \to 0$ leading to the simple relation $\mathbf{r}(t) = \phi(\mathbf{h}(t))$ used in the main text.

The estimator $\hat{\mathbf{x}}(t)$ of the latent state $\mathbf{x}(t)$ is obtained by a linear projection of the decoder rates $\mathbf{r}(t)$ onto the $M$-dimensional space of the latent dynamics

$$\hat{\mathbf{x}}(t) = W\mathbf{r}(t) = \frac{1}{N} \sum_{\alpha=1}^{M} \mathbf{w}_\alpha^T \mathbf{r}(t), \tag{S3}$$

Here, $\mathbf{w}_\alpha \in \mathbb{R}^N$ are the linear readout vectors, or coding directions. Each element in $\mathbf{w}_\alpha$ is drawn i.i.d. from the same distribution $\mathcal{P}(w)$. The coding directions are approximately orthogonal in the thermodynamic (large-$N$) limit, and the overlap between two different vectors is $\sum_i w_{\alpha i} w_{\beta_i} = O(1/\sqrt{N})$ for $\alpha \neq \beta$. The distribution is normalized so that $\mathbf{w}_\alpha^T \mathbf{w}_\alpha = N$. It follows that in the large $N$ limit $W^T W = \mathbf{I}$, where here $\mathbf{I}$ is the $M \times M$ identity matrix. Together, the readout vectors span the $M$ dimensional subspace of the latent dynamics.

Following the paradigm of predictive coding [3], we want the internal state of the network to represent the error in estimation. The error vector, or deviation of the current estimate from the target latent state is $\mathbf{x}(t) - \hat{\mathbf{x}}(t)$. We thus define internal state variables, that we identify as the membrane potential of the neurons, which are equal to the projection of the error into the $N$-dimensional neural space

$$\mathbf{h}(t) = b \left[ W^T(\mathbf{x}(t) - \hat{\mathbf{x}}(t)) \right]. \tag{S4}$$

Here, we have introduced an gain factor $b \in \mathbb{R}$, that defines the scale of the membrane potentials relative to the real readout error. We will soon identify this factor as the degree of balance in the network. We would like the dynamics of the network to have a stable attractor around $\mathbf{h} = 0$. The outputs of the neurons are a nonlinear transformations of the membrane potentials, so the output of neuron $i$ is given by $\phi(h_i(t))$.

In the analysis of the autoencoder in the main text, the dynamical equations of the membrane potentials $\mathbf{h}(t)$ were given *a-priori* by a the canonical circuit equations [4]. Here, on the other hand, the temporal evolution of $\mathbf{h}(t)$ is not independent, and is tied to the dynamics of the signal $\mathbf{x}(t)$ and of the readout $\hat{\mathbf{x}}(t) = W\mathbf{r}(t)$. To see how the membrane potentials evolve with time, we take a temporal derivative in both sides of (S4), which yields

$$
\begin{aligned}
\dot{\mathbf{h}}(t) &= b \left[ W^T (\dot{\mathbf{x}}(t) - \dot{\hat{\mathbf{x}}}(t)) \right] \\
&= b \left[ W^T \left( \frac{1}{\tau_x} A\mathbf{x}(t) + \frac{1}{\tau_x} \mathbf{x}_0(t) - W\dot{\mathbf{r}}(t) \right) \right] \\
&= b \left[ W^T \left( \frac{1}{\tau_x} AW\mathbf{r}(t) + \frac{1}{\tau_x} \mathbf{x}_0(t) + \frac{1}{\tau_x} W\mathbf{r}(t) - \frac{1}{\tau_x} W\phi\left(\mathbf{h}(t)\right) \right) \right]
\end{aligned}
$$

In the third step above we have used an approximation $\mathbf{x} \approx \hat{\mathbf{x}}(t)$; this approximation is valid as long as the readout error is small, and it introduces an error of $O(1/\sqrt{N})$ relative to the other $O(1)$ terms. Rearranging the terms, and absorbing the time constant $\tau_x$ within the free parameter $b$ we can write rewrite the dynamics as

$$
\dot{\mathbf{h}}(t) = b \left[ W^T \mathbf{x}_0(t) - W^T W \phi\left(\mathbf{h}(t)\right) + \Omega \mathbf{r}(t) \right]. \tag{S5}
$$

The first term is the input projected through feedforward weights. The second term is the inhibitory feedback implementing the error correction as we have seen in the main text. The last term is a recurrent term with weights given by $\Omega \equiv W^T(A + I)W$, where $I$ is the $N \times M$ identity matrix. Importantly, this feedback term is proportional to the decoder rates $\mathbf{r}(t)$, and not the output of the neurons $\phi(\mathbf{h})$. This is the term that implements the dynamics of $\mathbf{x}(t)$. It can be readily understood as it is the only term that contains the dynamic transfer matrix $A$ and the additional time constant, which is implicit inside the filtered readout $\mathbf{r}(t)$. In [1] they refer to these synapses as *slow* synapses, as they inherit the slow dynamics of the readout $\mathbf{r}(t)$. In general, this term is a temporal filter of the neural outputs, that introduces a longer time scale required to implement the encoded dynamical system.

Finally, to arrive at the full circuit equations analogous to (1) in the main text, we introduce membrane leak, weight disorder, added Gaussian noise, and delays:

$$
\tau \dot{h}_i(t) = -h_i(t) + \sum_j J_{ij} \phi(h_j(t - d)) + I_i(\mathbf{x}_0(t)) + b\Omega \mathbf{r}(t) + \sigma \xi_i(t), \tag{S6}
$$

where

$$
J_{ij} = g\mathcal{J}_{ij} - \frac{b}{N} \sum_{\alpha=1}^{M} w_{\alpha i} w_{\alpha j}, \qquad \text{and} \qquad I_i(\mathbf{x}_0(t)) = b \sum_{\alpha=1}^{M} w_{\alpha i} x_{0\alpha}(t). \tag{S7}
$$

Once again, we have absorbed a factor of $N$ within the arbitrary control factor $b$. The delays, noise, weight disorder, and leak are not part of the derivation, but can be seen as external constraints on the network. With the addition of the noise and disorder, we can naturally see the role of balance in the dynamics. It sets the effective scale of the error relative to the other driving forces in the network, which are the noise $\sigma$, and the emergent fluctuations due to the disorder, which are proportional to $g$. The mean-field derivation in the main text shows how the magnitude of $b$ affects the different sources of fluctuations in the network.

A noticeable difference from the mean-field equations for the autoencoder, is the added term $b\Omega\mathbf{r}(t)$, that in general changes the result of the mean-field derivation. However, in the limit where the latent dynamics is much slower than the membrane time constant, and both $\tau_x, \tau_r \ll \tau$, then the fluctuations in the decoder rates $\delta\mathbf{r}(t) = \mathbf{r}(t) - \langle \mathbf{r} \rangle$ are small and do not contribute to the overall fluctuations $\delta u(t)$ in (8). On the other hand, the contribution of the mean rates $b\Omega\langle\mathbf{r}\rangle$ will affect the bias in general.

We note that the linear dynamics can be generalized to nonlinear dynamics, by explicitly introducing nonlinearity within the readout (S2), and adapting the recurrent weights $\Omega$ accordingly. Similar ideas have been previously introduced in [5].

Finally, to get the autoencoder network studied in the main text, we can choose $A = -I$, yielding $\mathbf{x}(t) = \mathbf{x}_0$. With this choice of $A$, $\Omega = 0$, and the "slow" recurrent connectivity term in Equation (S6) drops out, leading to the circuit equations introduced in (1), only for $M$-dimensional signals instead of a scalar input. In the following section, we derive the full mean-field theory for an autoencoder for an input signal of $M$ dimensions, where $1 < M \ll N$.

## 2 Mean field theory for multidimensional stimuli

In section (3) of the main text, we calculate the variance of fluctuations in a scalar readout; here we generalize the calculation of variance to multidimensional stimuli by continuing from Equations (S6) and (S7). We will consider the more simple case of a network with no weight disorder and no delay, $g = d = 0$. Futhermore, as in the main text we consider an autoencoder without internal signal dynamics, i.e., $A = -I$ and $\Omega = 0$.

We define the readout in the direction $\alpha = 1, \ldots, M$ as

$$\hat{x}_\alpha(t) = \frac{1}{N} \mathbf{w}_\alpha^T \mathbf{r}(t). \tag{S8}$$

The dynamical fluctuations in the readout are given by

$$\delta \hat{x}_\alpha(t) = \hat{x}_\alpha(t) - \langle \hat{x}_\alpha \rangle. \tag{S9}$$

The readout error is determined by the bias and the variance of the readout. Below we show that the fluctuations in the readout $\delta \hat{x}(t)$ in each direction $\alpha = 1, \ldots, M$ are independent and so the total error can be written as

$$\varepsilon = \sqrt{\sum_\alpha^M (x_\alpha - \langle \hat{x}_\alpha \rangle)^2 + \sum_\alpha^M \langle \delta \hat{x}_\alpha^2 \rangle}, \tag{S10}$$

where the first term in the square root is the contribution of the bias, and the second term is the contribution of the variance of dynamical fluctuations.

### 2.1 First-order mean field theory for the bias

Following the same mean-field analysis as in the main text, we decompose the membrane voltage vector $\mathbf{h}$ into two contributions

$$\mathbf{h}(t) = \sum_{\alpha=1}^M \mathbf{h}_\alpha^\parallel(t) + \mathbf{h}^\perp(t). \tag{S11}$$

Here $\mathbf{h}_\alpha^\parallel = \boldsymbol{\mathcal{P}}_\alpha \mathbf{h}$ and $\mathbf{h}^\perp = (\mathbf{I} - \sum_{\alpha=1}^M \boldsymbol{\mathcal{P}}_\alpha)\mathbf{h}$ are the projections of the membrane potential vector onto the subspace spanned by $\{\mathbf{w}_\alpha\}$ and to the orthogonal subspace respectively. $\boldsymbol{\mathcal{P}}_\alpha = \frac{1}{N} \mathbf{w}_\alpha \mathbf{w}_\alpha^T$ is the orthogonal projection operator. Importantly, the readout vectors $\mathbf{w}_\alpha$ are approximately orthogonal in the large $N$ limit, enabling this decomposition. Thus $\mathbf{h}_\alpha^\parallel(t) = u_\alpha(t)\mathbf{w}_\alpha$ where $u_\alpha(t) \equiv \frac{1}{N} \sum_{i=1}^N w_{\alpha i} h_i(t)$. The two dynamical equations (4) in the main paper generalize to $M$ equations for the projections of the membrane potentials onto the subspace spanned by the readout vectors

$$\tau \dot{u}_\alpha(t) = -u_\alpha(t) + b[x_\alpha - \hat{x}_\alpha(t)] + \sigma \xi_\alpha^\parallel(t), \tag{S12}$$

and for the fluctuations in the orthogonal subspace

$$\tau \dot{h}_i^\perp(t) = -h_i^\perp(t) + \sigma \xi_i^\perp(t). \tag{S13}$$

The noise terms $\xi_\alpha^\parallel = \frac{1}{N} \sum_{i=1}^N w_{\alpha i} \xi_i$ reflect the projection of the single-neuron independent noise terms into the $\alpha$ readout direction, and $\xi_i^\perp$ is the independent noise in neuron $i$ in the orthogonal subspace. Since $M \ll N$, we can write, as in the main text, $\langle (\xi_i^\perp)^2 \rangle = \sigma^2$ and $(\xi_\alpha^\parallel)^2 = \frac{\sigma^2}{N}$.

Additionally, since the coding directions $\mathbf{w}_\alpha$ are approximately orthogonal, the $\xi_\alpha^\parallel$ are independent and $\langle \xi_\alpha^\parallel \xi_\beta^\parallel \rangle = 0$ for every pair $\alpha \neq \beta$.

The membrane potentials in the subspace orthogonal to all the coding directions, $\mathbf{h}^\perp(t)$ follow a simple OU process, and the variance of their fluctuations is given by $\langle (h_i^\perp)^2 \rangle = \frac{\sigma^2}{2\tau}$. Since the fluctuations in *all* readout directions $\delta u_\alpha(t)$ are small in the large $N$ limit, we can expand the activity of each neuron to linear order in these fluctuations

$$\phi(h_i(t)) = \phi\left(\sum_{\beta=1}^M w_{\beta i}\langle u_\beta \rangle + h_i^\perp(t)\right) + \phi'\left(\sum_{\beta=1}^M w_{\beta i}\langle u_\beta \rangle + h_i^\perp(t)\right)\sum_{\beta=1}^M w_{\beta i}\delta u_\beta(t). \quad \text{(S14)}$$

The decoder $\hat{x}_\alpha(t) = N^{-1}\sum_i w_{\alpha i}\phi(h_i(t))$ then reads

$$\hat{x}_\alpha(t) = \frac{1}{N}\sum_{i=1}^N w_{\alpha i}\phi\left(\sum_{\beta=1}^M w_{\beta i}\langle u_\beta \rangle + h_i^\perp(t)\right)$$
$$+ \frac{1}{N}\sum_{i=1}^N w_{\alpha i}\phi'\left(\sum_{\beta=1}^M w_{\beta i}\langle u_\beta \rangle + h_i^\perp(t)\right)\sum_{\beta=1}^M w_{\beta i}\delta u_\beta(t). \quad \text{(S15)}$$

Mirroring the derivation in the main paper, we take a temporal average of the decoder, and use $\langle \hat{x}_\alpha \rangle = x_\alpha - \langle u_\alpha \rangle / b$, to obtain a set of $M$ self-consistent equations for the order parameters $\langle u_\alpha \rangle$,

$$x_\alpha - \frac{\langle u_\alpha \rangle}{b} = \int \mathcal{D}z \prod_{\beta=1}^M (dw_\beta \mathcal{P}(w_\beta))\, w_\alpha \phi\left(\sum_{\beta=1}^M w_\beta \langle u_\beta \rangle + \frac{\sigma}{\sqrt{2\tau}}z\right). \quad \text{(S16)}$$

These equations can be solved numerically to give the stationary solutions for $\langle u_\alpha \rangle$. In the main text we highlight an intuitive graphical solution. While the basic idea is similar in the multidimensional setting, the graphical solution is less intuitive since the LHS of (S16) is a nonlinear integral equation involving all of the order parameters $\langle u_\alpha \rangle$.

In the mean-field solution the bias is the Euclidean distance between $\langle \hat{\mathbf{x}} \rangle$ and $\mathbf{x}$, given by

$$\varepsilon_{bias} = \frac{1}{b}\sqrt{\sum_\alpha^M \langle u_\alpha \rangle^2}. \quad \text{(S17)}$$

The bias $u_\alpha(\mathbf{x}, \sigma)/b$ is a function of the noise and the inputs $x_\alpha$ in all directions $\alpha = 1, \ldots, M$. This is a deterministic function that can be inverted by, for example, training of an efferent readout which can eliminate the error due to bias. In the next section, we calculate the error due to dynamical fluctuations in the different coding directions. These depend on the noise and chaos in the network and are not easily removed by a static readout.

## 2.2 Mean-field theory for the second order statistics of the fluctuations

We now turn to study the fluctuations around the static first-order mean-field solution. By removing the time average from the expansion in (S15), we identify the fluctuations in the readout as

$$\delta\hat{x}_\alpha(t) = \frac{1}{N}\sum_{i=1}^N w_{\alpha i}\phi'\left(\sum_{\beta=1}^M w_{\beta i}\langle u_\beta \rangle + h_i^\perp(t)\right)\sum_{\beta=1}^M w_{\beta i}\delta u_\beta(t). \quad \text{(S18)}$$

As we have noted above, the coding directions are all random and $\frac{1}{N}\sum_i w_{\alpha i}w_{\beta i} = \delta_{\alpha\beta}$. As a result the fluctuations $\delta u_\alpha(t)$ in different directions decouple and follow the linear dynamics

$$\tau\delta\dot{u}_\alpha(t) = -\delta u_\alpha(t) - b\langle \phi' \rangle_\alpha \delta u_\alpha(t) + \sigma\xi_\alpha^\parallel(t). \quad \text{(S19)}$$

Here, the average $\langle \phi' \rangle$ is performed the statistics of the stationary solution calculated above, and depends in the means $\langle u_\alpha \rangle$ in all $M$ directions,

$$\langle \phi' \rangle_\alpha = \int \mathcal{D}z \prod_{\beta=1}^{M} (dw_\beta \, \mathcal{P}(w_\beta)) w_\alpha^2 \phi' \left( \sum_{\beta=1}^{M} w_\beta \langle u_\beta \rangle + \frac{\sigma}{\sqrt{2\tau}} z \right). \tag{S20}$$

Since fluctuations decouple, we can solve the equation in each direction $\alpha$ independently, and the fluctuations in each direction are given by

$$\langle (\delta u_\alpha(t))^2 \rangle = \frac{\sigma^2}{2\tau N(1 + b\langle \phi' \rangle_\alpha)}, \qquad \text{and} \qquad \langle (\delta \hat{x}_\alpha(t))^2 \rangle = \frac{\langle \phi' \rangle_\alpha^2 \sigma^2}{2\tau N(1 + b\langle \phi' \rangle_\alpha)}. \tag{S21}$$

Finally, since the fluctuations are orthogonal and independent, the total contribution of the fluctuations to the readout error is given by $\sqrt{\Delta}$, where $\Delta$ is variance of the decoder across all readout directions

$$\Delta = \sum_{\alpha}^{M} \langle (\delta \hat{x}_\alpha(t))^2 \rangle. \tag{S22}$$

## 3 Dynamic mean-field theory for balanced networks with weight disorder

In section 4 of the main text we study the effect of weight disorder and deterministic chaos on the error, and show how balance suppress the fluctuations at the readout. Here, we present with more details the mean-field solutions for chaotic networks with synaptic balance, and the approximations we introduced in order to study the effects of the balance on the dynamics. For simplicity, we derive the solutions here assuming a scalar input signal, as introduced in the main text. Furthermore, for notational brevity we set the membrane time constant to be $\tau = 1$.

First, we note that the first-order mean-field solution for the bias (5) is unaffected by the dynamics of the noise, and is similar whether the fluctuations of the membrane potential arise from deterministic chaos or from additive Gaussian noise. However, the mean-field solution requires averaging over the membrane potential fluctuations in the directions orthogonal to the readout, which in general may be different in the case of deterministic chaos. In the case of additive Gaussian noise, we have shown that the temporal average of the fluctuations are $\langle \delta(h_i^\perp)^2 \rangle = \sigma^2/2$ for all $i$. When the fluctuations are the result of deterministic chaos, the variance $\langle (h_i^\perp)^2 \rangle$ is found self consistently via Dynamic Mean Field Theory (DMFT) [6]. In the following section, we highlight the main ideas in deriving DMFT for the emergent fluctuations in the membrane potential.

### 3.1 Dynamic mean-field solution for the fluctuations in the orthogonal subspace

We now turn to compute the statistics of the fluctuations of a random network in its chaotic phase, when the variance of the weight distribution is above the critical transition point $g > g_c$. The dynamic mean field theory for a chaotic neural network was first introduced by [6] and re-derived later by [7, 8, 9, 10, 11, 12]. The connectivity in the subspace orthogonal to the readout direction is randomly distributed, thus the properties of the fluctuations in this subspace, $\delta h^\perp(t)$, are equivalent to previous studies of random neural networks. We bring the highlights here, and refer the reader to [7] for a more detailed account of the derivation.

We define the autocorrelation function of the chaotic fluctuations as

$$\Delta^\perp(s) \equiv \frac{1}{N} \sum_i \left\langle \delta h_i^\perp(t) \, \delta h_i^\perp(t+s) \right\rangle \tag{S23}$$

where, as before, $\delta \mathbf{h}^\perp(t) = (I - \mathcal{P})\delta \mathbf{h}(t)$. The variance of the fluctuations is given by the equal-time autocorrelation $\Delta^\perp(0)$. In DMFT, the autocorrelation is obtained by properly averaging over the dynamic equation for the fluctuations $\delta h_i^\perp(t)$, given by the equation in the RHS of (4). The result, is a second-order differential equation for $\Delta^\perp(s)$ given by

$$\left(1 - \frac{\partial^2}{\partial s^2}\right) \Delta^\perp(s) = g^2 q(s). \tag{S24}$$

Here on the LHS we have a second-order differential operator acting on the autocorrelations of the membrane potential. On the RHS, we have the autocorrelation function of the fluctuations in the firing rates of the neurons $\phi_i(t) \equiv \phi(h_i(t))$, is given by

$$q(s) = \frac{1}{N} \sum_i \langle \delta\phi_i(t)\delta\phi_i(t+s)\rangle. \tag{S25}$$

Here $\delta\phi_i(t) = \phi_i(t) - \langle\phi_i\rangle$ are the temporal fluctuations in the output of neuron $i$ about its mean firing rate $\langle\phi_i\rangle$. The mean autocorrelation of the firing rates $q(s)$ is given by taking a statistical average over the weight disorder in the system, and can be written as

$$q(s) = \int Dz \left( \int Dy \phi(\sqrt{\Delta^\perp(0) - \Delta^\perp(s)}y + \sqrt{\Delta^\perp(s)}z) \right)^2. \tag{S26}$$

Plugging (S26) into (S24) we get a self-consistent integro-differential equation for $\Delta^\perp(t)$. The boundary conditions for this equation are given by $\dot{\Delta}^\perp(s) = 0$ and $\dot{\Delta}^\perp(\infty) = 0$, corresponding to the smoothness of the autocorrelation at $s = 0$ and the conditions for the existence of a chaotic solution at $s = \infty$ respectively. The solution can be found by numerically evaluating the second order differential equation [7]. The variance of the fluctuations in the orthogonal subspace $N^{-1}\sum_i \langle \delta h_i^{\perp 2}\rangle = \Delta^\perp(0)$ is used in the static solutions $\langle\phi\rangle$ and $\langle\phi'\rangle$ above.

## 3.2 Dynamic mean-field for the fluctuations in the readout direction

The dynamics of the fluctuations in the direction of the readout is given by

$$\tau\delta\dot{u}(t) = -\beta\delta u(t) + g\eta^\parallel(t), \tag{S27}$$

where $\beta \equiv 1 + b\langle\phi'\rangle$. Here, the noise term $\eta^\parallel(t)$ reflects the projection of the recurrent feedback $\mathcal{J}\phi$ with random connectivity $\mathcal{J}$ onto the coding direction $\mathbf{w}$. The mean of the recurrent noise $\eta^\parallel(t)$ is given by

$$\langle\eta^\parallel(t)\rangle = \frac{1}{N}\sum_{ij} w_i \mathcal{J}_{ij}\langle\phi(h_i(t))\rangle = \frac{a_\mathcal{J}}{\sqrt{N}}\langle\phi\rangle, \tag{S28}$$

where $a_\mathcal{J} \sim \mathcal{N}(0,1)$ is a random number drawn from the standard normal distribution. The random number depends on the particular realization of $\mathcal{J}$ and readout vector $\mathbf{w}$, and does not vanish in the large $N$ limit. Requiring *detailed-balance* in the disordered connectivity, i.e., the constraint $\sum_j \mathcal{J}_{ij} = 0$, $\forall i$ can remove this bias term. Without detailed-balanced weights, the realization-specific temporal mean needs be incorporated within the mean-field equation for $u$, and will generally add to the bias error. We note that the expected $a_\mathcal{J}$ across different readout directions is zero. As we argued before, the static bias can be removed by an efferent readout. However, in the case of weight disorder, the bias term is random and depends on the actual realization of $\mathcal{J}$, and there is no analytical solution for the bias. Nevertheless, the static bias can be easily removed by training the linear readout. The bias correction to the mean-field is needed even for the dynamical phase below the chaotic transition, $g < g_c$.

For networks in the chaotic phase, we must also consider the temporal fluctuations. The autocorrelation of the noise term in (S27) is given by

$$\langle\delta\eta^\parallel(t)\delta\eta^\parallel(t')\rangle = \frac{1}{N^2}\sum_{ijkl} w_i w_j \mathcal{J}_{ik}\mathcal{J}_{jl}\langle\delta\phi_k(t)\delta\phi_l(t')\rangle = \frac{1}{N}q(t-t') + O(1/N^2) \tag{S29}$$

where $q(t-t')$ is the mean autocorrelation of the outputs given in (S26), which can be found self-consistently as highlighted above. Unlike the mean, it is self-averaging, and does not depend on the specific realization of $\mathcal{J}$ in the large-$N$ limit.

To find an expression for the autocorrelation function

$$\Delta(s) = \langle\delta u(t)\delta u(t+s)\rangle, \tag{S30}$$

we follow the same logic as when deriving Dynamic Mean-Field Theory for the fluctuations $\Delta^\perp(s)$ above [7]. First, we take the Fourier transform of the dynamical equations (S27) for the fluctuations in the readout direction $\delta u(t)$:

$$(-i\omega - 1 - \beta)\delta\tilde{u}(\omega) = g\eta^\parallel(\omega). \tag{S31}$$

Next, we multiply the expression by its complex conjugate and take another Fourier transformation back to the temporal representation. Replacing the variance of $\eta^\parallel\rangle$ with the variance of the recurrent connectivity (S29) we obtain

$$\left((1 + b\langle\phi'\rangle)^2 - \frac{\partial^2}{\partial s^2}\right)\Delta(s) = g^2 q(s). \tag{S32}$$

The boundary conditions on the second-order differential equation are, as above in equation (S24), are $\dot{\Delta}(0) = \dot{\Delta}(\infty) = 0$. The full solution for $\Delta(s)$ can be evaluated numerically using the solution for $q(s)$ describes in the previous section. However, to get further insight into how the solution behaves with $b$, we would like to derive an analytical expression. In the following section, we approximate the chaotic autocorrelation function with a more simple model with a colored Gaussian noise term that permits analytical treatment.

### 3.3 Approximating the chaotic fluctuations with temporally colored Gaussian noise

The exact temporal correlation function of the chaotic fluctuations is complicated, and depends on the details of the problem, such as the nonlinearity, sources of noise and the external input [7, 8]. However, it has some common characteristics: (1) it is a symmetric function $q(s) = q(-s)$; this is due to time reversal symmetry in the system. (2) It is an exponentially decaying function; this is because the chaotic dynamics is characterized by a positive Lyapunov exponent [13]. (3) The decay time is of the order of the membrane potential, which is the only time scale in the network. The last point is true away from the critical transition point $g = g_c$, where critical slowing down can result in long-range temporal correlations [7]. The exact shape of the autocorrelation however, depends on the details of the problem. For example it may be convex or concave, depending on external noise sources [8].

While the detailed function is not analytically tractable in many cases, we can replace the chaotic fluctuations with a more simple noise model that captures the important aspects of the chaotic fluctuations, namely, symmetric and exponentially decaying with time constant similar to the membrane time constant. We write the dynamics of the fluctuations in the direction of the readout in (S27) as

$$\tau\dot{\delta u}(t) = -\beta\delta u(t) + g\zeta(t), \tag{S33}$$

where $\beta = 1 + b\langle\phi'\rangle$. Here, we have replaced the chaotic fluctuations in the coding direction, $\eta^g(t)$ with correlated Gaussian noise $\zeta(t)$ with zero mean and autocorrelation function given by

$$\langle\zeta(t)\zeta(t+s)\rangle = \frac{1}{N}\exp\left(-\frac{|s|}{2\tau}\right). \tag{S34}$$

This noise can be easily realized with a filtered white Gaussian noise

$$\tau\dot{\zeta}(t) = -\zeta(t) + \frac{1}{\sqrt{N}}\zeta'(t), \tag{S35}$$

where $\langle\zeta'(t)\zeta'(t')\rangle = \delta(t - t')$. For brevity of notation, in the following we will set membrane time constant $\tau = 1$.

In (S33) we have a stochastic ODE with a corresponding to a particle undergoing gradient descent in a deterministic quadratic potential, but driven by colored noise. If at time $t_0$ the location of the particle is known, then the variance in the location of the particle at time $t$ is given by [14]

$$\alpha(t_0, t) = 2\int_t^{t'} ds\,\exp[-2\beta(t_0 - s)]\int_{t_0}^s dr\,C(s - r)\exp[-\beta(s - r)]. \tag{S36}$$

Here $C(s - r)$ is the autocorrelation function of the colored Gaussian driving noise, and is given by $C(s - r) = \frac{1}{N} \exp[-(s - r)/2]$ for $s > r$. We thus obtain

$$
\begin{aligned}
\alpha(t_0, t) &= \frac{2}{N} \int_{t_0}^{t} ds \exp[-2\beta(t - s)] \int_{t_0}^{s} dr \exp[-(\beta + \frac{1}{2})(s - r)] \\
&= \frac{2}{N(\beta + \frac{1}{2})} \int_{t_0}^{t} ds \exp[-2\beta(t - s)] \left( 1 - \exp[-(\beta + \frac{1}{2})(s - t_0)] \right) \\
&= \frac{\exp[-2\beta t]}{N(\beta + \frac{1}{2})} \int_{t_0}^{t} ds \left( \exp[2\beta s] - \exp[(\beta + \frac{1}{2})t_0] \exp[(\beta - \frac{1}{2})s] \right). \quad \text{(S37)}
\end{aligned}
$$

If the balance is strong, we can write $\beta = b\langle \phi' \rangle \gg 1$. This small approximation allows us to simplify the above expression by ignoring $O(1)$ corrections to $\beta$, and write

$$
\begin{aligned}
\alpha(t_0, t) &= \frac{\exp[-2\beta t]}{N\beta} \int_{t_0}^{t} ds \left( \exp[2\beta s] - \exp[\beta t_0] \exp[\beta s] \right) \\
&= \frac{\exp[-2\beta t]}{N\beta^2} \left( \frac{1}{2} \left( e^{2\beta t} - e^{2\beta t_0} \right) - e^{\beta t_0} \left( e^{\beta t} - e^{\beta t_0} \right) \right) \\
&= \frac{1}{N\beta^2} \left( \frac{1}{2} \left( 1 - e^{-2\beta(t - t_0)} \right) - e^{\beta t_0} \left( e^{-\beta t} - e^{\beta t_0 - 2\beta t} \right) \right) \\
&= \frac{1}{2N\beta^2} \left( \left( 1 - e^{-2\beta(t - t_0)} \right) - 2 \left( e^{-\beta(t - t_0)} - e^{-2\beta(t - t_0)} \right) \right). \quad \text{(S38)}
\end{aligned}
$$

As mentioned above, if we interpret the ODE as the motion of a particle in a quadratic potential driven by colored noise, then $\alpha(t_0, t)$ denotes the variance in the location of the particle at time $t$, if the location is known at time $t_0$. In that case the variance in the location of the particle at the *steady state* is given by setting $t_0 = 0$ and $t = \infty$, yielding

$$
\alpha(0, \infty) = \frac{1}{N2\beta^2} = \frac{1}{2N\tilde{b}^2} = \frac{1}{2N\langle \phi' \rangle^2 b^2}. \quad \text{(S39)}
$$

Here we have again used $\tilde{b} \gg 1$ so $\beta = 1 + \langle \phi' \rangle b \approx \langle \phi' \rangle b$.

Using the result for the variance at the steady state, we obtain an expression for the fluctuations in $\delta u(t)$

$$
\langle \delta u^2 \rangle \approx \frac{g^2}{2\tilde{b}^2 N}. \quad \text{(S40)}
$$

Finally, the fluctuations of the readout $\hat{x}(t)$ are given by

$$
\langle \delta \hat{x}^2 \rangle \approx \frac{\langle \phi' \rangle^2 g^2}{2\tilde{b}^2 N}. \quad \text{(S41)}
$$

The average over the steady state $\langle \phi' \rangle$ is solved using the mean-field equations, using the variance $\Delta^{\perp}(0)$ found above using dynamic mean-field theory.

## 4  Delays, noise and resonance

In this section, we study the response of a balanced network with delayed feedback to an external white noise. We begin by considering the characteristic equation of the delayed ODE in (10) in the absence of noise,

$$
G(z) = z\tau + 1 + \tilde{b}e^{-zD} = 0. \quad \text{(S42)}
$$

The real and imaginary parts of the complex number $z = \gamma + i\omega$ represent the exponential growth and oscillations of the solution ansatz. As discussed in the main text, below the critical balance $\tilde{b}_c$, all solutions to this equation have negative real part $\gamma < 0$. In this regime the dynamics is stable and the fluctuations decay to zero rapidly.

In the presence of noise, the system is constantly driven. The autocorrelation function of the fluctuations in this state is defined as $\Delta(s) = \langle \delta u(t)\, \delta u(t+s) \rangle$. To study the response of of $\delta u(t)$ to the external noise, we look at the Fourier components of the autocorrelation function

$$\hat{\Delta}(\omega) = \frac{1}{2\pi} \int ds\, e^{i\omega s} \Delta(s). \tag{S43}$$

In the model driven by white noise, the integrated power across all frequencies is $\sigma^2/2N$. The power at a specific frequency $\omega$ is given by

$$\hat{\Delta}(\omega) = \frac{\sigma^2}{2NG(i\omega)G^*(i\omega)} = \frac{\sigma^2}{2N(i\omega\tau + 1 + \tilde{b}e^{-i\omega D})(-i\omega\tau + 1 + \tilde{b}e^{i\omega D})}. \tag{S44}$$

Using the characteristic equation we know that $G(\omega_c) = 0$ when the balance is $\tilde{b} = \tilde{b}_c$. Plugging this equality into (S44), we obtain an expression for the response of the network at the resonant frequency $\omega_c$,

$$\hat{\Delta}(\omega_c) = \frac{\sigma^2}{2N(\tilde{b}_c - \tilde{b})^2}. \tag{S45}$$

We approximate the total contribution to the fluctuations as the sum of fluctuations in the absence of delay plus the contribution of the resonance in $\omega_c$, yielding

$$\hat{\Delta}(\omega) = \frac{\sigma^2}{2N} \left( \frac{1}{(1 - \tilde{b}_c)^2 + \omega^2} + \frac{1}{(\tilde{b}_c - \tilde{b})^2 + (\omega - \omega_c)^2} \right). \tag{S46}$$

Finally, using the Wiener–Khinchin theorem we can find the total variance of the fluctuations, which is given by

$$\Delta = \int d\omega \hat{\Delta}(\omega). \tag{S47}$$

Plugging (S46) in (S47) and integrating, we arrive at eq. (12).

Finally, we note that for the chaotic network, the derivation would be similar, only the variance of the white noise $\sigma^2/N$ is replaced with $\hat{q}(\omega)/N$, which is the Fourier representation of the rate autocorrelation in (S26) . Since $q(s)$ is exponentially decaying, we have $\hat{q}(\omega) \ll 1$ for $\omega \gg 1/\tau$. In the case of small delays $d \ll \tau$ the the noise at the critical frequency $\hat{q}(\omega_c) \ll 1$ and thus the resonant effects in this case are negligible.