[Reviews · NeurIPS 2020]

Review 1

Summary and Contributions: This paper studies the coding accuracy of networks with 'tight' or 'classic' balance within a mean-field theory. The new formulation allows to dissociate the different factors influencing network performance (type of balance, delays, weight disorder, and noise), and is simple enough to allow for in-depth mathematical analysis. The latter is used to analyze several network properties and behaviors such as the expected bias and variance of a coded signal, and the interactions of balance, noise (or chaos) and delays.

Strengths: Overall, the paper is well written, and I think it presents an elegant unification of various properties of balanced networks in a mathematically tractable way. Some of the results had been known before individually, and this paper sheds new light on them.

Weaknesses: A potential weakness of the paper is that it seeks to analyze the properties of balanced *spiking* networks, but does so in a firing rate framework. That leaves the question of how the presented firing-rate models relate to the previous spiking models (beyond a reference to an anonymized paper, which I can therefore not judge). These differences should be more clearly elaborated. Also, the paper only shows results for static 1D representations. While the supplementary material provides theoretical results for extensions (that seem sound), it would nice if these were demonstrated and compared to numerical results.

Correctness: Everything seems correct

Clarity: The paper is generally well written. A few suggestions for improvement: (1) Figure 1 - there is too much variance in font-sizes in the figure (2) Figure 2 - the example traces and sampling of parameters could use more than one 'oscillations' example. (3) All figures - I appreciate that the paper should fit in 8 pages, but the figures and axis labels etc. are made so small as to be hard to read in many places. (4) Line 38 - When discussing predictive coding, a reference is made to Eliasmith & Anderson (2004), which does not seem to be the correct reference.

Relation to Prior Work: Yes

Reproducibility: Yes

Additional Feedback:


Review 2

Summary and Contributions: This paper addresses the theoretical principles governing the efficacy of balanced predictive coding and its robustness to noise, synaptic weight heterogeneity and communication delays. For this purpose, the paper introduces an analytically tractable model of balanced predictive coding, in which the degree of balance and the degree of weight disorder can be dissociated. This model is used thereafter to infer a mean field theory of coding accuracy. The work is a step forward towards dissecting the differential contributions of neural noise, synaptic disorder, chaos synaptic delays, and balance, to the fidelity of predictive neural codes.

Strengths: The mathematical formulation seems to be sound, and the results relevant.

Weaknesses: The decoding of the input used in the main body of the paper is a strongly simplifying assumption, that largely ignores the sophisticated nonlinear transformation of the network itself.

Correctness: The claims made by the paper seem to be correct.

Clarity: The paper is clearly written, but it may be hard to read for researchers not familiar with the theoretical work in spiking neural networks domain.

Relation to Prior Work: The previous work seems to be well addressed.

Reproducibility: Yes

Additional Feedback:


Review 3

Summary and Contributions: The goal of this paper is to understand under which conditions recurrent networks efficiently encode stimuli. To this end, the manuscript focuses on a class of models that interpolate between randomly-connected and predictive coding networks. Using a detailed mean-field theory, the authors determine the influence of different parameters (recurrent feedback, noise, random connectivity, delays) on stimulus encoding. The theoretical predictions show an excellent match to simulations, and provide a clear picture of how various parameters influence coding.

Strengths: - remarkable theoretical analysis - unified picture for a variety of different modelling frameworks

Weaknesses: The paper ends up not answering the main question set up in the Abstract and Intro. The initial goal is to determine the conditions under which the coding error is 1/N rather than 1/sqrt(N), where N is the size of the network. But most of the paper focuses on the scaling with other parameters, in particular the feedback b. The scaling with N is not shown in the Figures, nor mentioned in the Discussion (although it can be directly inferred from the results). This looks like an important omission, but can be fixed easily. My understanding is that 1/N scaling can never be achieved in presence of delays? Update: unfortunately the last question was not answered in the authors' response.

Correctness: yes

Clarity: Very clearly written.

Relation to Prior Work: The model in absence of delays, and much of the corresponding mean-field theory, look like a specific case of the class of models studied in Ref 23. It would be fair to state this. This fact does not remove anything to the merits of the present study. The main questions addressed here (negative feedback and its impact on coding) were not treated in Ref 23. Discussion: it may be important to comment on the fact that Ref 7 ascribed the 1/N scaling to the spiking nature of the networks. One possible criticism of the present work is that it deals with rate rather than spiking networks. I don't think the conclusions would change in spiking networks, but it may be useful to pre-empt this criticism.

Reproducibility: Yes

Additional Feedback: Minor comments: l. 87: "were" -> "where" l.128: the relation to E-I balanced networks could be made more explicit. In some versions of those networks, there are also two independent effective parameters that scale separately the negative feedback and the variance of the connectivity (see e.g. Mastrogiuseppe and Ostojic 2017) l. 223 "the full solution for the chaotic system is highly involved" - the solution for adiabatic inputs seems to be available from Ref.23, but perhaps the situation here is different? My understanding is that we are here in the adiabatic limit, not in the case of Ref 38? In the adiabatic case, why does the (finite) correlation timescale of the noise matter for coding? Is there a transition out of chaos as either b or the strength of the stimulus are increased? It would be useful to clarify these points.


Review 4

Summary and Contributions: This work theoretically analyzed the dynamics of neural networks achieving predictive coding in balanced conditions, and gives us insight into understanding how different factors including noise, synaptic disorder, synaptic delay, and balance affect the network performance. It is a vluable piece of work to the field. The authors partly addressed my concerns, and I keep the same score.

Strengths: The theoretical analysis is quite deep. The concept of balanced predicitive coding was proposed already, but this study gives us some new insight on details.

Weaknesses: The analysis is based on a simple form of the network dynamics, Eqs.1-2, which seems to be quite artificial, e.g., the feedforward input takes the form of w_ix, and the same w_i appears in the recurrent connections and the read-out vector. It is unclear to me how a real neural system achieves this. Can the authors give an example?

Correctness: The claims and method in this paper are reasonable.

Clarity: It is OK, consider the authors have to compress many mathematical details into 8 pages.

Relation to Prior Work: It is clearly stated.

Reproducibility: Yes

Additional Feedback:

[Author Response · NeurIPS 2020]

**Overall:** We thank the reviewers for their comments and questions. We are encouraged to see that all reviewers recommended to accept the paper. The one repeated criticism is the simplicity of the model we study, and in particular the use of a firing-rate based dynamics rather than spiking neurons. It is an understandable question, in particular when the framework we present is conceptually similar to previous studies on efficient coding of spiking networks [Boerlin *et al.*, 2013]. When considering the work on spiking networks, it is tempting to think that the superclassical efficiency, namely the scaling of the readout error as $1/N$, is a consequence of the spiking dynamics. Instead we consider a more *general* model and assert that the efficiency is the result of the strong negative feedback that scales as $b \sim N$. Furthermore, our theory is agnostic towards the local nonlinear transfer function. Thus, we believe that the simplicity of the model is rather a strength, as it strips the framework only to the necessary components. It is a fair question to ask if our results hold for spiking dynamics as well. As a first approximation, one can consider a spiking neuron as a renewal process with Poisson statistics where the Poisson mean is the instantaneous firing rate and is given by $\phi(x_i(t))$. The transfer function here could be the mean-field firing-rate approximation for the spiking model. Here, it is easy to be convinced that the spiking dynamics is equivalent to an extra noise term [Kadmon & Sompolinsky, 2015]. As mentioned in the paper, numerical simulations of the efficient coding framework with Leaky Integrate and Fire (LIF) neuron show similar qualitative results [Chalk *et al.*, 2016]; these observation were not understood theoretically. We note that we have derived a rigorous theory for LIF Neurons. However, the spiking dynamics require a different mean-filed approach which is out of the scope of the current paper. We will publish this theory elsewhere as it does not undermine the novelty nor the significance of the more general results we present here.

In the following we address other concerns and questions raised by the reviewers.

**Reviewer 1:** In addition to the concern on the rate-based dynamics, which we have addressed above, the reviewer noted that we did not include numerical simulation for time-dependent and high-dimensional signals. The treatment of these was delegated to the supplementary material (SM), together with the encoding of autonomous dynamical systems as they do not provide additional insights. However, we agree with the reviewer that we can add a figure demonstrating the theory for high dimensional dynamic signals to the SM. On the camera ready version of the paper we have slightly more space, and we can increase the size of figures and fonts to make them more legible, following the reviewers comments. The references on predictive coding will be also corrected.

**Reviewer 2:** The main goal of our theory is to understand how well a large network of unreliable units can faithfully represent a signal. We did not attempt to claim that the simple setting we use preforms complex computations. It is actually a well known criticism of Balance Network, that they can represent only linear transformation [Ahmadian & Miller, 2019]. However, it does not mean that the network cannot perform more complex computations. In the SM we show how the network can implement a general autonomous linear dynamical system. In [Alemi *et al.*, 2017] a similar framework was used to encode nonlinear autonomous dynamics. Finally, a recent work has shown that Balanced Networks can implement rectified-linear transformations and be used to perform a variety of nonlinear computations [Baker, Zhu & Rosenbaum, 2020]. Nevertheless, the question of how the computation can be reliably encoded by the population is ubiquitous, and that is the problem we address in this work. Following the reviewers comment, we will emphasize in the discussion the relevance of our work for nonlinear computations.

**Reviewer 3:** We thank the reviewer for noting that we did not emphasize enough the difference in scaling of the readout error. In particular, a central point is that in the case of tight balance [Deneve & Machens, 2016], the negative feedback scales as $b \sim N$. We emphasize that previous studies achieve superclassical scaling of error $(1/N)$ because of the scaling of feedback. We believe that showing that the superclassical error can be obtained in rate-based network with arbitrary nonlinear neurons is rather a strength of our theory.

The reviewer mentioned that previous works have decoupled the random connectivity from the ordered part (e.g. [Mastrogiuseppe & Ostojic, 2018]). However, they did not use them in a strong balance setting as a mean to decouple the random fluctuations from the magnitude of structured part. We will elucidate our approach in the text. Lastly, the reviewer asks about the difficulties of the full chaotic solution. Indeed, the autocorrelation of the chaotic fluctuation has been calculated before by several authors. However, to solve for the fluctuations of the *readout*, the autocorrelation of the chaotic fluctuations is used as a noise source in the ODE for the decoder in Eq. (9). An exact solution for the chaotic autocorrelation is usually found numerically by solving the mean-field equations. Thus, the exact expression for the readout error can only be evaluated numerically. In order to gain insight into the solution, we propose a simple approximation that captures the correct error scaling, and reveals the mechanism by which the negative feedback efficiently suppress chaotic fluctuations, namely the low-pass filter on the fluctuations.

**Reviewer 4:** The reviewer correctly notes feedforward input weights and recurrent connectivity are closely tied to ensure the balanced cancellation of feedforward input by the recurrent connectivity as in many predictive coding frameworks like [Deneve & Machens, 2016]. Hebbian learning of recurrent synapses driven by feedforward weights $\mathbf{w}$ could learn the recurrent part $\mathbf{w}\mathbf{w^T}$. Any errors in the learned weights are accounted for by the random part $\mathcal{J}$.

[Meta-Review · NeurIPS 2020]

This paper theoretically investigates and experimentally verifies properties of predictive coding network. Although limited to simple representations, the mathematical analysis has impressed the reviewers. The AC thus recommends acceptance of this work.